# Human Fall Detection Based on Body Posture Spatio-Temporal Evolution

**DOI:** 10.3390/s20030946

**Published:** 2020-02-10

**Authors:** Jin Zhang, Cheng Wu, Yiming Wang

**Affiliations:** School of Rail Transportation, Soochow University, Suzhou 215011, China; zhangjin1983@suda.edu.cn (J.Z.);

**Keywords:** computer vision, fall behavior detection, five-point inverted pendulum model, motion instability, human posture spatio-temporal map, rotational energy

## Abstract

Abnormal falls in public places have significant safety hazards and can easily lead to serious consequences, such as trampling by people. Vision-driven fall event detection has the huge advantage of being non-invasive. However, in actual scenes, the fall behavior is rich in diversity, resulting in strong instability in detection. Based on the study of the stability of human body dynamics, the article proposes a new model of human posture representation of fall behavior, called the “five-point inverted pendulum model”, and uses an improved two-branch multi-stage convolutional neural network (M-CNN) to extract and construct the inverted pendulum structure of human posture in real-world complex scenes. Furthermore, we consider the continuity of the fall event in time series, use multimedia analytics to observe the time series changes of human inverted pendulum structure, and construct a spatio-temporal evolution map of human posture movement. Finally, based on the integrated results of computer vision and multimedia analytics, we reveal the visual characteristics of the spatio-temporal evolution of human posture under the potentially unstable state, and explore two key features of human fall behavior: motion rotational energy and generalized force of motion. The experimental results in actual scenes show that the method has strong robustness, wide universality, and high detection accuracy.

## 1. Introduction

Understanding and recognizing human behavior remains a challenging and important task in the field of computer vision. It can be applied to a variety of research cases, such as intelligent video surveillance, human–computer interaction and video content understanding. According to the different recognition targets, it can be divided into two categories: human normal behavior recognition and classification [1,2,3,4,5,6,7,8] and human abnormal behavior detection and warning [9,10,11,12,13]. With the development of research theories and methods, human normal behavior recognition and classification mainly study how to distinguish the different behaviors of human beings, and it has evolved into research on the breadth and diversity of different behaviors. However, human body abnormal behavior detection and early warning focuses on the study of the existence of abnormalities in human behavior sequences, and continues to develop research into the detection depth and scene complexity of a single behavior.

There are many different types of abnormal behaviors in the human body, and there are different definitions. Common abnormal behaviors often include both group behavior and individual behavior. Typical group anomalies include crowding, gathering and fighting, etc. Typical individual abnormal behaviors include falls, retrogrades and evasion, etc. In crowded public places, an individual’s abnormal fall due to unknown reasons may lead to more vicious group abnormal events, such as crowding, panic, and even trampling. In particular, if an elderly person or child who lacks mobility has not been discovered and treated in time after falling, it is likely to cause more harm and even death. Our research can help staff detect the behaviors in real time, automatically and intelligently in a large amount of video data from the video surveillance system and make correct emergency responses in time. It can play a positive role in the intelligent development of public safety. Therefore, the effective detection of human fall behavior has important theoretical research value and social significance in public scenes.

However, falls in public places are hard to find. There are two well-known difficult issues here. One is in public places, and various uncontrollable disturbances often appear randomly in the scene. The other is that the fall behavior itself has a different direction. When the view of camera and the human body fall in the same line or direction, it is called the posture of same direction (SD posture), the morphology change in the fall process is not obvious. These two problems inevitably lead to a significant decline in the robustness of identifying human abnormal fall behavior in the wild environment. In order to cope with the two problems above, we study the intrinsic dynamics of human fall behavior from the human body structure. By explaining the morphological evolution characteristics of the human body’s Part Affinity Fields (PAFs) in the unstable process, a motion description model based on spatio-temporal information coupling is established. The abnormal fall behavior detection method proposed in this article can dynamically analyze and judge motion continuously, with high precision and strong robustness, which is very suitable for fall recognition in complex scenes. In this article, our contributions are:We propose a five-point inverted pendulum model based on the key points of the human body structure to describe the human motion pose. Based on this model, we construct a human body posture spatio-temporal evolution map to reflect the pose evolution process in a time series.Using the theory of human body dynamics, two quantitative characteristics of rotational energy and generalized force of motion are proposed, and the general instability law of fall behavior is studied. Our method is able to detect fall behavior in different directions and achieve higher precision.

The rest of this article is organized as follows. Section 2 describes the related work in the detection of human fall behavior. Section 3 gives our system model, the five-point inverted pendulum model, and describes its spatio-temporal information coupling mechanism. Section 4 proposes a fall detection method based on the above model. We performed a thorough performance evaluation in Section 5. Finally, Section 6 concludes our work.

## 2. Related Work

The existing fall detection methods based on machine vision have been widely studied. The flow of the algorithm is generally composed of three stages: image preprocessing, feature extraction, classification, and detection. The implementation and optimization of the latter two stages have become the focus of the research.

From the perspective of feature extraction, existing solutions can be divided into two categories: static features and dynamic features. Static feature analysis is essentially a morphological model of fall behavior [14,15,16,17,18,19,20]. Typical examples are the shape matching cost defined by the full Procrustes distance [14] and the application of an approximate ellipse [19] to describe the shape of human body. The human contour is simplified as three basic parts of the head, body and legs, and at the same time, the orientation, sum of the heights and height ratio are calculated and used to analyze the shape of the human [17,21]. Then, a statistically determined method is used to collect a set of ratios of the human body part occupied area of each frame and the writer uses it as input data of the MEWMA chart [18,19,20,22,23]. In the above methods, their performance must depend on the extraction integrity of the contour of the human body. However, in the actual scene, it is difficult for the target human body to avoid being blocked, which may affect the detection effect. In addition, some features are only sensitive to partial fall posture recognition, and, in most cases, it is difficult to detect such on SD posture. In summary, there are too many limitations in the application of this type of method.

Compared to statical features, dynamic features are more concerned about change of motion. Generally, abnormal events that are different from the global motion trend can be identified through motion speed and direction. The trajectory of the detected moving object is clustered and an anomaly detection model is defined in [24]. By detecting motion path [25] or the change of energy [26], anomalies can be reflected. Ref. [27] proposes a novel efficient high-level representation of trajectories called snapped trajectories, and combines object trajectory analysis and pixel based analysis for abnormal behavior inference. This enables detecting abnormal behaviors related to speed and direction of object trajectories, as well as complex behaviors related to finer motion of each object.

In recent research, dynamic features can more fully describe the changes in the continuity of anomalous events by combining spatial and temporal features. Most abnormal behaviors may last for a while from beginning to end. We can not judge the result only by the limited features in a single frame. These methods first construct feature vectors based on spatial feature extraction, and then apply time analysis theory to improve their performance. This is a sequential and decomposed fusion method. However, other studies have introduced the use of spatial and temporal features to construct transient fusion feature vectors. Typically, Ref. [28] defines a new feature called Cmotion, which quantitatively describes the intensity of of motion on spatio-temporality. Ref. [1] proposes a new structure called Space Time Graph Convolutional Network (ST-GCN) for motion identification. A large number of experimental evaluations have demonstrated that these transient fusion feature vectors have significant advantages and can break through the limitations of previous methods by automatically learning spatial and temporal patterns from the data.

From the perspective of classification and detection, recent research has evolved from simple threshold determination to complex intelligent machine learning. This process illustrates that feature selection and application are shifting from low-level feature discrimination to advanced feature learning. According to the modeling requirements of anomaly classification, supervised learning algorithms are usually used, such as hidden Markov model [28], neural network [18], and single or multi-layer support vector machines [27]. In addition to traditional machine learning methods, the deep learning classification model has become a new research hotspot. Convolutional neural networks (CNN) with deep structural features have been widely studied and applied, such as [1,3,5], and their improved algorithms, Region-CNN (R-CNN), fast R-CNN, Faster-RCNN [29] etc.

At present, in the published research, the detecting fall method of focusing on human body structure is still rare. However, in fact, the dynamic characteristics of the human body pose structure in the motion process are the fundamental basis for judging abnormal falls. Ref. [2] has also been able to detect PAFs of human bone structure more stably through deep learning models. Based on the Part Affinity Fields (PAFs) of the human skeleton structure, it has great research value to find the key points closely related to the abnormal fall of the human body and to establish the coupling model of time and space information at these key points in the fall process. Therefore, in our work, we focus on extracting and studying the essential characteristics of abnormal fall behavior and discussing the relationship between human body structure and posture stability. By quantitative analyzing spatio-temporal characteristics of fall motion, we propose posture stability theory based on video analysis and reveal the general rules of falls in different directions in two-dimensional (2D) images. Our method to detect abnormal fall behavior is more adaptive in real scenes.

## 3. System Modelling

In the public scenes such as subway stations, bus stops and streets, the human behavior types are different from in indoor scenes. The normal behaviors of people are generally walking, standing or sitting. Fall event is a dynamic behavior, so static behavior of sitting is ignored. Then, the normal behaviors mainly include walking and standing. We mainly distinguish fall behavior from them. Falling behavior is a direct result of the loss of stability during human movement. Whether it is due to external forces or due to physical internal causes, the body has a dynamic change process from the height to the ground. Compared with the normal walking and standing, the movement and morphological changes of the fall process have significant characteristics. The key points of the human skeleton help to deeply explore the motion essential characteristics of the fall process. Therefore, this article proposes the analysis and detection of the fall anomaly behavior based on the coupled map of motion space-time information. Starting from the human motion dynamics, the physical characteristics and quantization algorithms of the loss of stability are studied, and a series of continuous multi-scenario multi-directional falls are collected and tested to prove its rationality and universality for the detection of abnormal behavior. The study consists of four steps, as shown in Figure 1:Introducing two-branch multi-stage CNN to extract the Part Affinity Fields of human skeleton structure and establishing a description model of human body instability—five-point inverted pendulum model;Based on the five-point inverted pendulum model, constructing the vectors of human body motion using spatio-temporal information coupling;Using the principle of human body dynamics and analyzing the characteristics of human instability caused by abnormal behavior of falling and its quantitative theory;Realizing the detection and identification of human fall behavior.

### 3.1. Five-Point Inverted Pendulum Model for Human Body Motion Instability

The stability of motion is defined as the nature of an object or system that returns to its original motion after it has deviated from its motion by external disturbances. If the object or system can gradually return to the original motion, the motion is stable, otherwise it tends to be unstable. The human body can be considered to be in a limited stable equilibrium when walking or standing. When the human body is disturbed by external force or is unable to return due to the body’s own reasons, the fall will occur, and the process is understood as motion instability.

In order to analyze the motion instability of the human body, we must first establish a description model that fully describes the motion posture and is as simple as possible. The significance of the simplification is to reduce the computational complexity and improve the robustness in a multi-person scene. We start from the classic human standing balance model: the inverted pendulum model [30]. This model is widely used in robot motion balance and stability research [30,31,32,33]. According to the theory of human body dynamics, the balance and stability of the human body are closely related to the position of the human body and the position of the support plate. Therefore, the face and upper limbs are ignored, and the trunk parts, including the head, the neck, the buttocks, and the left and right knees that reflect the support position to a certain extent, are retained, shown in Figure 2. Thus, the key points of the inverted pendulum of the body movement instability include: head (*H*); neck (*N*); buttocks (*B*); left knee ($K_{l}$); right knee ($K_{r}$). Then, according to the natural connection rules of the human body, (H,N), (N,B), $\left(B,K_{l}\right)$, $\left(B,K_{r}\right)$ are connected, and $\left(K_{l},K_{r}\right)$ are connected according to the inverted pendulum shape, thereby constructing the five-point inverted pendulum model for the movement instability process of the human body. The model can not only effectively reflect the changes of the posture of the movement, but also relatively simplify the structure of the human body, and avoid interference with the analysis of the posture due to the heterogeneity and randomness of multiple key points of the body.

### 3.2. Spatio-Temporal Coupling Body Posture Evolution Based on an Inverted Pendulum Model

The human fall behavior is a dynamic process that changes from a finite steady state to an unstable state. Through a large number of observations on the instability of the movement, the human fall behavior has the following characteristics:The beginning of the fall behavior has fuzzy uncertainty: the human body may be unstable due to external force interference, or may be out of equilibrium due to the body’s own reasons;The fall behavior has time-varying and continuity: after the fall behavior occurs, the human body’s posture changes dynamically, and each part of the body continuously changes its position according to its own degree of freedom;The posture of fall behavior is random, while the overall trend of the posture has a commonality: the trend of fall motion is constantly close to the ground, and the center of gravity of the body is reduced. However, due to the human stress reaction and the surrounding environment, the behavioral posture is different in detail;The end of the fall behavior has static stability: the end of the fall behavior is often the human body on the ground which can be regarded as static in a short time.

According to the characteristics of the above-mentioned fall behavior, the recognition of fall behavior must not rely solely on the body contour at a certain moment. It is extremely necessary to construct the evolution of human poses with coupling the information on time and spatial dimension. Based on the proposed five-point inverted pendulum model, we present the following steps for construction:Within each frame, we extract the morphological features of the human body’s PAFs, and establish a transient human body description map according to the “five-point inverted pendulum” model;In two adjacent or equally spaced frames, we connect the same key points on the inverted pendulum to form a motion vector capable of reflecting the displacement of the inverted pendulum;In the continuous video frames of the specified duration, the human inverted pendulums on the adjacent video frames are connected in pairs, and a motion vector set embodying the evolution timing of the human body posture is constructed, that is, the temporal and spatial information is coupled with the human body posture evolution map;In the time-space information coupling human body posture evolution map, we solidify the key point of each inverted pendulum *B*, and construct a standardized time-space information to couple the human body posture evolution map.

In the standard evolution map obtained above, we naturally retain the spatial position information of the key points of the human body motion at the complete time sequence, and make the motion trajectory of the key points of the human body pose in the form of vectors, as shown in Figure 3.

A standard evolution map based on the five-point inverted pendulum model is able to show a series of positions for each individual key point. Particularly, we define the standard evolution map using a directed graph.

**Definition** **1.**
*A human body posture spatio-temporal evolution map is a directed graph on a series of body skeletons with N key points (here, N is set to 5) in T frames featuring both inter-body and intra-frame connections:*
G=(V,E).

*$V=\{v_{i}^{t}\}$—the node set including the all N key points of body skeletons in T continuous frames, where i denotes the i-th key points having $i\in \{H,N,B,K_{l},K_{R}\}$ and t denotes the t-th frame having t∈{1,2,⋯,T};*

*$E=\{E_{s},E_{f}\}$—the edge set including two subsets:*
-
*$E_{s}$ denotes the connections between the key points in an individual body skeleton at one frame having $E_{s}=\left\{\left(v_{i}^{\left(\cdot \right)},v_{j}^{\left(\cdot \right)}\right)\right\}$, where i and j are in $\left\{H,N,B,K_{l},K_{R}\right\}$ and i≠j;*
-
*$E_{f}$ denotes the connections between the same key points in consecutive or equally spaced frames having $E_{f}=\left\{\left(v_{\left(\cdot \right)}^{t},v_{\left(\cdot \right)}^{t+1}\right)\right\}$, where t and t+1 are two consecutive frames.*




With the definition, all edges in $E_{f}$ represent the trajectory of one particular key point over time. If there are multiple persons in the scene, we need to construct the spatio-temporal evolution maps of human body using $G_{p}={\left(V,E\right)}_{p}$, where $p\in \{P_{1},P_{2},P_{3},\ldots \}$.

## 4. Human Fall Detection Method Based on Motion Instability

Based on the five-point inverted pendulum model, the human body posture spatio-temporal evolution map uses motion vector trajectory to couple the dynamic process of the spatial position change of the human body’s PAFs in continuous time. The evolution map of abnormal falls and normal walking is shown in Figure 3. From the figure, we can easily find that there is a significant difference on the motion vector trajectory between abnormal fall and normal walking. Now, we need to analyze the difference and extract robust features that can be used for quantitative analysis.

### 4.1. Energy Analysis of Human Body Particle System Based on Moment of Inertia

In our proposed spatio-temporal evolution map of human body posture, in order to standardize, the key point *B* of the five-point inverted pendulum model is solidified. The objective motivation for fixing the key point *B* is that (1) The *B* on the hip part is less affected by human body rotation on the *z*-axis, which is perpendicular to the image plane; (2) The impact of the human body’s height difference can be reduced to some extent. In the real three-dimensional world, when the fall occurs, the human body’s five-point inverted pendulum forwardly or backwardly swings about $90^{\circ }$ with *B* as the center. In the two-dimensional image, when the fall occurs, the five-point inverted pendulum of the human body appears to rotate at the *z*-axis only. When walking, it does not have the characteristics of rotational motion. Therefore, the fixed-axis rotational energy is introduced for quantification. The value of rotational energy is higher, and the motion instability is stronger.

The moment of inertia I is a measure of the inertia of the fixed axis rotation of a rigid body, which is defined as Equation (Equation 1):(1)$$I=\sum_{E\;P_{i}}I_{i}=\sum_{E\;P_{i}}m_{i}r_{i}^{2},$$
where $m_{i}$ is the mass of point ${E\;P}_{i}$, and $r_{i}$ is the distance from the point ${E\;P}_{i}$ to the rotation axis. Figure 4 gives a simple particle system composed of two particles of the end point $E\;P_{1}$ and end point $E\;P_{2}$. To explain clearly, five-point inverted pendulum model with a different color shows the state of motion at different moments. The yellow shows state of motion in the *t*-th frame and the blue shows state of motion in the (t+1)-th frame. The rotation energy $E_{rot}$ is given in Equation (Equation 2):(2)$$E_{rot}=\frac{1}{2}\sum_{E\;P_{i}}I_{i}\omega_{i}^{2},$$
where $\omega_{i}$ is the angular velocity of the point ${E\;P}_{i}$. Assume that each end point is unit mass, where $m_{i}=1$.

Next, we adopt the rotation energy $E_{rot}$ as the feature to recognize the existence of fall behavior. Positive samples are different directional falling events and negative samples include standing and walking in public places. Each rotation energy $E_{rot}$ is extracted from these samples. In addition, for the comprehensive study on the fall behavior in different directions posture, we arranged the experimenters to fall in the eight directions at the same position in the No. 3 scene of Table 1. The spatio-temporal evolution maps for falling and walking in the eight directions are shown in Figure 5. From the maps, we can see that there are some significant differences between two behaviors. The results using the rotational energy based on moment of inertia demonstrate the feasibility of analyzing fully the motion instability.

The pseudo code of the fall detection algorithm is generalized and shown in Algorithm 1.



### 4.2. Generalized Force Analysis of Human Fall Based on Lagrangian Mechanics

The five-point inverted pendulum of the human body can be seen as a complex connecting rod system. In order to study the human body fall process, we need to describe the posture and motion state of the human body’s five-point inverted pendulum, which must analyze its dynamic characteristics. A similar study is mainly embodied in the dynamics of robots [30,31,33]. The method is to establish the dynamic model of the robot by applying the Lagrangian mechanical equation. The basis of Lagrangian mechanics is the variational calculation of system energy from system variables and time. As the complexity of the system increases, the Lagrangian mechanics of motion can become relatively simple.

The human body fall based on the five-point inverted pendulum of the human body is mainly based on the rotational motion in Lagrangian mechanics. The model is excavated to dynamic characteristics of the body movement change and a two-connecting rod system is constructed, as shown in Figure 6. The key point *B* of the system is taken as the origin of the system. The process in our method is as follows:Select (H,N) and (N,B) to construct a two-connecting rod system;Select the coordinate system and the complete and independent generalized variable connecting rods $R\;D_{1}$ and $R\;D_{2}$;Find the kinetic energy $K_{R\;D_{1}}$ and $K_{R\;D_{2}}$, and potential energy $P_{R\;D_{1}}$ and $P_{R\;D_{2}}$ of each component of the robot, and construct a Lagrangian function;Calculate the generalized force $Q_{R\;D_{1}}$ and $Q_{R\;D_{2}}$ on the corresponding connecting rods;Substitute the Lagrange equation to obtain the dynamic equation of the robot system.

The dynamic equations are established in the two-connecting rod system. We denote that the mass of connecting rod $R\;D_{1}$ is $m_{R\;D_{1}}$, and $K_{R\;D_{1}}$ represents the kinetic energy of the rod $R\;D_{1}$. $K_{R\;D_{1}}$ can be calculated using Equation (Equation 3):(3)$$K_{R\;D_{1}}=\frac{1}{2}m_{1}{\upsilon_{1}}^{2}.$$

Here,
(4)$${\upsilon_{1}}^{2}={\dot{x}}_{1}^{2}+{\dot{y}}_{1}^{2}.$$

Consider
(5)$$\left\{\begin{array}{c} x_{1}=-\left(d_{1}\sin\left(\theta_{1}+\theta_{2}\right)+d_{2}\sin\theta_{2}\right)\\ y_{1}=-\left(d_{1}\cos\left(\theta_{1}+\theta_{2}\right)+d_{2}\cos\theta_{2}\right) \end{array}\right.$$
we have:(6)$$\left\{\begin{array}{c} {\dot{x}}_{1}=-\left(d_{1}\cos\left(\theta_{1}+\theta_{2}\right)\left({\dot{\theta }}_{1}+{\dot{\theta }}_{2}\right)+d_{2}\cos\theta_{2}{\dot{\theta }}_{2}\right)\\ {\dot{y}}_{1}=d_{1}\sin\left(\theta_{1}+\theta_{2}\right)\left({\dot{\theta }}_{1}+{\dot{\theta }}_{2}\right)+d_{2}\sin\theta_{2}{\dot{\theta }}_{2} \end{array}\right.$$

At the time, $K_{R\;D_{1}}$ can be calculated as follows:(7)$$\begin{array}{cc} K_{R\;D_{1}} & =\frac{1}{2}m_{1}\left[d_{2}^{2}{\dot{\theta }}_{2}^{2}+d_{1}^{2}\left({\dot{\theta }}_{1}^{2}+2{\dot{\theta }}_{1}{\dot{\theta }}_{2}+{\dot{\theta }}_{2}^{2}\right)\right.\\ & +2d_{1}d_{2}\cos\theta_{1}\left({\dot{\theta }}_{1}{\dot{\theta }}_{2}+{\dot{\theta }}_{2}^{2}\right)] \end{array}$$

We also denote $P_{R\;D_{1}}$ as representing the potential energy of the rod $R\;D_{1}$. We then have:(8)$$\begin{array}{cc} P_{R\;D_{1}} & =m_{1}g\left(y_{3}-y_{1}\right)\\ & =m_{1}g\left(d_{2}\cos\theta_{2}+d_{1}\cos\left(\theta_{1}+\theta_{2}\right)\right). \end{array}$$

Similarly, the mass of connecting rod $R\;D_{2}$ is $m_{R\;D_{2}}$. $K_{R\;D_{2}}$ represents the kinetic energy of the rod $R\;D_{2}$, which is:(9)$$K_{R\;D_{2}}=\frac{1}{2}m_{2}{\upsilon_{2}}^{2}=\frac{1}{2}m_{2}d_{2}^{2}{\dot{\theta }}_{2}^{2}$$

$P_{R\;D_{2}}$ represents the potential energy of the rod $R\;D_{2}$, which is:(10)$$P_{R\;D_{2}}=m_{2}gh_{2}=m_{2}gd_{2}\cos\theta_{2}$$

Thus, we can obtain the total kinetic energy *K* of the system and the total potential energy *P* of the system:(11)$$K=K_{R\;D_{1}}+K_{R\;D_{2}}$$
(12)$$P=P_{R\;D_{1}}+P_{R\;D_{2}}$$

According to Lagrange’s Equations, the Lagrangian *L* is defined as:(13)L=K−P

We then get the generalized force in the two-connecting rod joint space, $Q_{R\;D_{1}}$, and $Q_{R\;D_{1}}$ using Equations (Equation 14) and (Equation 15): (14)$$\begin{array}{cc} Q_{R\;D_{1}} & =\frac{d}{dt}\frac{\partial L}{\partial {\dot{\theta }}_{1}}-\frac{\partial L}{\partial \theta_{1}}\\ & =m_{1}d_{1}{\ddot{\theta }}_{1}^{2}\\ & +\left(m_{1}d_{1}^{2}+m_{1}d_{1}d_{2}\cos\theta_{1}\right){\ddot{\theta }}_{2}\\ & +m_{1}d_{1}d_{2}\sin\theta_{1}{\dot{\theta }}_{2}^{2}-m_{1}gd_{2}\sin\left(\theta_{1}+\theta_{2}\right)\;\;\;\;\;\;\;\;\;\; \end{array}$$
(15)$$\begin{array}{cc} Q_{R\;D_{2}} & =\frac{d}{dt}\frac{\partial L}{\partial {\dot{\theta }}_{2}}-\frac{\partial L}{\partial \theta_{2}}\\ & =\left(m_{1}d_{1}^{2}+m_{1}d_{1}d_{2}\cos\theta_{1}\right){\ddot{\theta }}_{1}\\ & +\left[\left(m_{1}+m_{2}\right)d_{2}^{2}+m_{1}d_{1}^{2}+2m_{1}d_{1}d_{2}\cos\theta_{1}\right]{\ddot{\theta }}_{2}\\ & -2m_{1}d_{1}d_{2}\sin\theta_{1}{\dot{\theta }}_{1}{\dot{\theta }}_{2}-m_{1}d_{1}d_{2}\sin\theta {\dot{\theta }}_{1}^{2}\\ & -\left(m_{1}+m_{2}\right)gd_{2}\sin\theta_{2}-m_{1}gd_{1}\sin\left(\theta_{1}+\theta_{2}\right) \end{array}$$

Thus, the generalized force Q can be derived using Equation (Equation 16):(16)$$Q=Q_{R\;D_{1}}+Q_{R\;D_{2}}$$

In the light of the theory of motion stability, if the generalized force Q meets the condition of $Q=Q_{R\;D_{1}}+Q_{R\;D_{2}}\to 0\;or\;constant$, the motion is balanced or in a stable state; otherwise, the motion basically tends to instability. In the standard spatio-temporal evolution map, there are *T* five-point inverted pendulum models in total. Their generalized forces $Q^{t}$ are analyzed sequentially; when the instability is accumulated to the state that it cannot be adjusted dynamically, it will almost lead to the fall behavior. This is the intrinsically dynamic principle of the fall behavior. Therefore, the pseudo code of the fall detection algorithm is generalized and shown in Algorithm 2.



### 4.3. The Spatio-Temporal Motion Feature Tensor Construction and Classification

Combining the two types of characteristic indicators of the above analysis, including the rotational energy and the combined motion force, the characteristic data of the “five-point inverted pendulum” model in the spatio-temporal map are coded according to the time series and PAFs, and a feature tensor is constructed using the time and space characteristic of the fall instability movement. The coding method is described as follows:In the *T* frame of video images, a person’s standard spatio-temporal evolution map G=(V,E) can construct a feature sequence. If there are multiple people in the scene, we need to build a standard spatio-temporal evolution map by humans $G_{p}={\left(V,E\right)}_{p}$, where $p\in \{P_{1},P_{2},P_{3},\ldots \}$, and repeat the same steps.We define three forms of new feature tensors, namely $E_{C\;O\;D\;E}$, $Q_{C\;O\;D\;E}$ and $F_{C\;O\;D\;E}$, where $E_{C\;O\;D\;E}$ consists only of rotational energy, $Q_{C\;O\;D\;E}$ consists only of generalized forces, and $F_{C\;O\;D\;E}$ is rotated energy and generalized forces are combined.In the *T* frame of images, we can calculate *T* results of $E_{rot}$ and arrange them in chronological order to form a feature tensor. The feature tensor is set to $E_{C\;O\;D\;E}$.In the *T* frame of video images, we can calculate *T* results of $Q^{t}$, which contain $Q_{1}^{t}$ and $Q_{2}^{t}$. We also arrange them in chronological order to form a feature tensor. This feature tensor is denoted as $Q_{C\;O\;D\;E}$, which is 2×T in length.We concatenate $Q_{C\;O\;D\;E}$ and $E_{C\;O\;D\;E}$ in order to construct a new feature tensor $F_{C\;O\;D\;E}$ with a length of (T−1+2×T)=3×T−1 (In the experiment, the frame rate of the video is 25 fps, the sampling frequency is 5 fps, and T is 6).

From this, we get the third algorithm for fall detection, whose pseudo code is shown in Algorithm 3. An example of $F_{C\;O\;D\;E}$ is shown in Table 2.



Based on the human skeleton results of deep learning architecture, further analysis, and research on the extracted body posture, we obtain more fundamental dynamic motion characteristics of the feature tensor. In the next step, we use a relatively simple but robust supervised learning method—the logistic regression model to form a binary classifier to detect and identify the fall behavior caused the human instability.

## 5. Experimental Evaluation

When detecting human abnormal behavior, “abnormalities” are often specified by the researchers and occur in some particular scenarios. Therefore, the baseline data lack a systematic and comprehensive test. In previous studies, such as a Hockey Fight in a stadium [34], Violence in Subway (ViS) [35] on a subway platform, video clips from CAVIAR project [36] and FallVideo [17] in indoor corridors and halls, the BEHAVE project’s dataset on outdoor roads (not crowded) [37], and Violent Flows [38], these scenarios are different and focus on their specific problems.

Our experimental dataset called “Postures of Fall (PoF)” is different from existing fall datasets, such as the University of Rzeszow fall detection dataset (URFD) and the fall detection dataset (FDD) [23] and Multi-cam [19] in the indoor scenes. The comparison of these datasets and the proposed dataset (PoF) is shown in Table 3. It is further highlighted that PoF serves the research of the the public scenes. In this article, we mainly study human fall detection in different directions, especially for SD posture. SD posture is the more challenging posture for the fall detection, as shown in sample images in Table 1 scenarios 1 and 2.

There are 10 scenarios in PoF dataset of about 486 M that contain many real-world challenges such as multiple people, occlusion, and contact. This dataset includes two parts: (1–4) lab-based and (5–10) real-world tests, as shown in Table 1. According to the proposed algorithms, the important precondition is establishing standard spatio-temporal evolution map: $G_{p}={\left(V,E\right)}_{p}$. In (1–4) lab-based scenarios, it is easy to finish as shown in Figure 3 and Figure 5. However, in real-world video from the video surveillance system, especially (8–10) scenarios, a standard spatio-temporal evolution map needs a more reliable extraction model of a human skeleton to ensure key points as complete as possible. The problem that key points are missing or a mistake is still inevitable. Considering the availability of the algorithm, the supplementary mechanism is as below:Five-point inverted pendulum model which is missing more than three key points will be considered invalid. The three key points need include head(*H*), neck (*N*), and buttocks (*B*). They are the critical conditions where the proposed method works.In the standard spatio-temporal evolution map, if the invalid five-point inverted pendulum model in *T* frames exists, the model of the former sample frame takes the place of the invalid one.In the multiple persons scenarios, there is the matching problem of a five-point inverted pendulum model for each motion individual in the consecutive sample frame. For each individual, we may have several candidates in the candidate region based on the effective motion distance. Inspired by [2], we score each candidate model using the reciprocal sum of Euclidean distance computation on the same type of key point, and the higher score is regarded to be stronger correlation between two models.

The extraction results of standard spatio-temporal evolution map in real-world scenarios as shown in Figure 7. In the processing, the problem of the missing or mistake key points still existed—for example, the shoulder part is wrongly detected as the head. However, standard spatio-temporal evolution maps showed in the right column are also very significant characteristic for the fall behavior. The feature analytical method is proved to be fault-tolerant and robust, and fit for the detection requirements of the real complex scenes.

The last step logistic regression classifier is used to identify abnormal fall behavior by the human motion feature tensors. Firstly, for in order to verify the feasibility of algorithms, the 1–4 scenes samples, 50% of them are used for training and the others used for testing. Then, we use ROC and AUC to test and analyze the performances of three algorithm classifiers, as shown in Figure 8.

The main difference between the three algorithms is the component of the feature tensor. We use the same dataset and logistic regression model parameters. The classifier performance indicators of the three algorithms are shown in Table 4. From the graph in Figure 8, the Area Under Curve (AUC) and Average Precision (AP) values of the third classifier are as high as about 99.4% and 96%, respectively, which reflects the reasonableness of classifier in Receiver Operating Characteristic (ROC) curves and the detection capability of fall behavior (positive samples) in P–R curves. Algorithm 1 (ALG1) and Algorithm 2 (ALG2) can detect abnormal fall behavior to a certain extent because the characteristics of rotational energy or generalized force can fully describe the human body posture and distinguish between falling and walking, as shown in Figure 9. The processed feature data are represented by a radar map in eight directional fall and walking samples. Compared with the first two algorithms of ALG1 and ALG2, the results fully demonstrate that the two fusion feature tensors are more efficient with Algorithm 3 (ALG3) with rotational energy $E_{rot}$ and generalized force Q.

In addition, Radviz analyzed sample data for three feature tensors (radial coordinate visualization). By mapping a multidimensional feature variable to a two-dimensional space by a nonlinear algorithm, the characteristics of the data pattern can be more clearly presented. Then, we get the same conclusion, that is, using the feature tensor $F_{C\;O\;D\;E}$ is more reasonable and effective than using the feature tensors Q and $E_{rot}$ alone, as shown in Figure 10. The feature tensor $F_{C\;O\;D\;E}$ consists of 17 variables and contains 12 generalized force features and five rotation energy characteristics. It has good classification results and also proves the rationality of the feature tensor coding rules.

Table 4 shows that the recall values of the positive samples in the three algorithms are not very satisfactory. This is because part of the algorithm is mainly based on kinetic energy, so there are some limitations in detecting slow-down behavior. According to the experimental analysis, it is poor at detecting abnormal fall samples in scenario 1, which has a negative impact on the performance of the fall test and also confirms this problem.

However, the outstanding advantage of the proposed algorithm, especially ALG3, is that the robustness of the detection is better at the level of medium (normal) motion intensity even under adverse conditions such as occlusion and contact. In addition, it appears to be able to effectively recognize SD posture that cannot be distinguished from the conventional morphological indicators due to being more confused with a walking posture. Next, in order to prove the superiority of the algorithm, we test ALG3 using FallVideo [23]. The FallVideo data set is available at http://foe.mmu.edu.my/digitalhome/FallVideo.zip. The results in Table 5 show that the proposed ALG3 results are better than the traditional feature analysis methods.

The final goal of this paper is to solve the detection of abnormal fall behavior in the public scenes, and we add a large amount of data from the video surveillance system into the PoF to further verify the performance of the algorithm. As shown in Table 1, No. 5–7 scenes are simple multi-person scenes, No. 8 scene is occluded multi-person scenes, and No. 9–10 are extremely challenging crowded scenes. The performance of fall detection is 95.1%, 88.7%, and 79.3%, respectively, by filtrating of invalid objects and the supplementary mechanism of modeling. In addition, the algorithm can assure the real-time ability in the sampling frequency 5 fps condition. It illustrates the value of the algorithm in practical application and provides reference for the development of intelligence of anomaly detection in outdoor public scenes.

## 6. Conclusions

Our focus is on detecting abnormal fall behavior caused by motion instability. The high robustness of human bone detection based on deep learning is the premise and basis of this research. We designed a simplified but very elaborate five-point inverted pendulum model for human body, and used the human dynamics theory to analyze and construct the human body posture spatio-temporal evolution map related to fall behavior. Through the spatio-temporal evolution map, we further analyze and describe the basic characteristics of human fall behavior, including the rotational energy and generalized force of motion, avoiding the difficulty of extracting the foreground or edge of motion in the real scene in the traditional methods, which theoretically guarantees the robustness of the algorithm. Our experiments not only use the mainstream video datasets but also provide a new datasets of fall postures in different directions. The experimental results show excellent accuracy and high robustness. It also proves that the proposed feature tensor can provide new research ideas and attempts abnormal fall detection. 

## Figures and Tables

**Figure 1 sensors-20-00946-f001:**
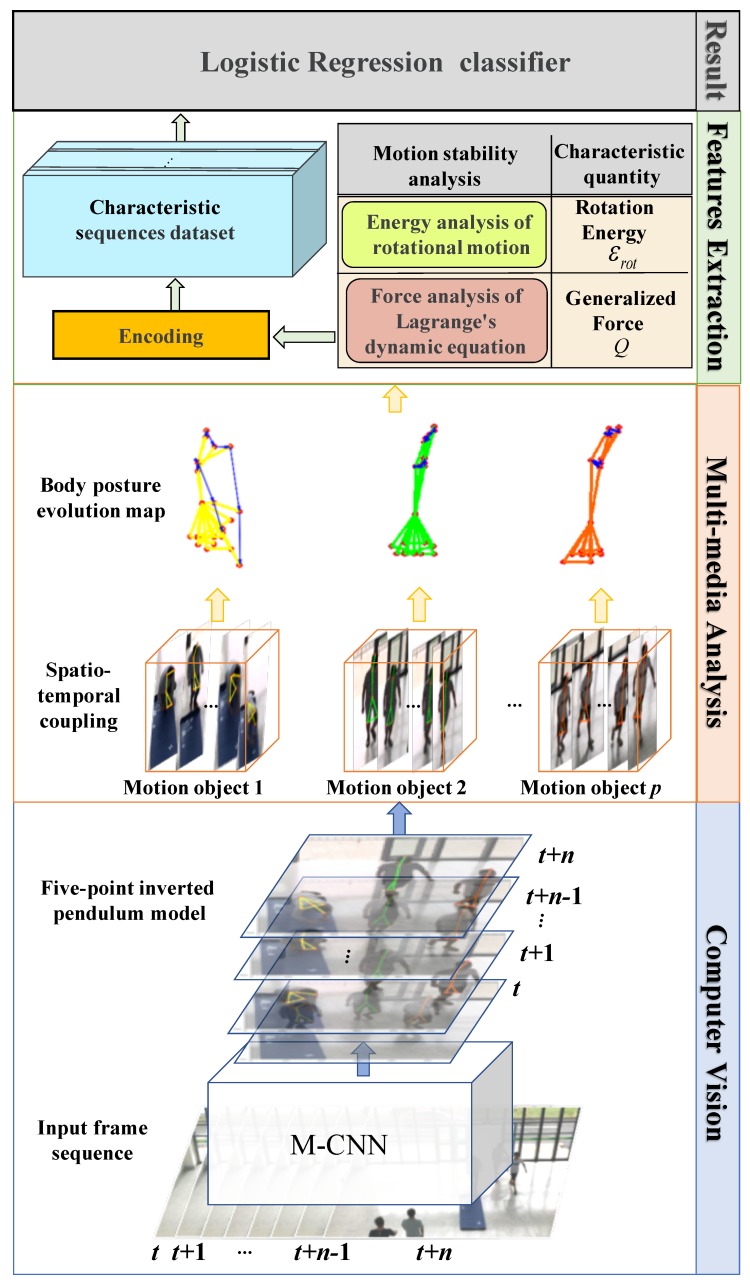
An overview of the proposed method for abnormal fall behavior detection.

**Figure 2 sensors-20-00946-f002:**
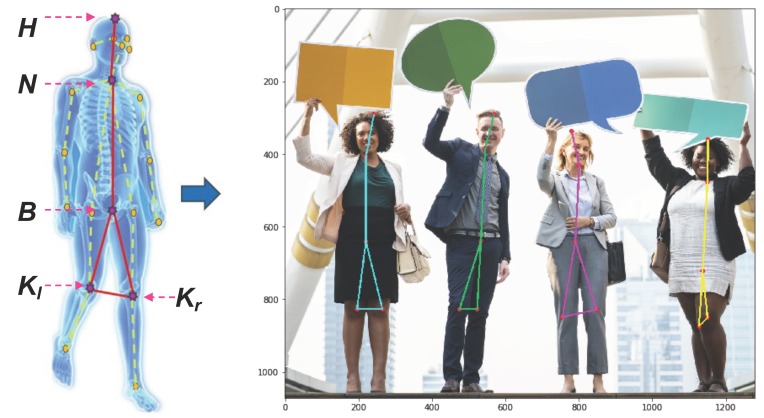
Five-point inverted pendulum model. The left model indicates COCO model [2] that includes 18 key parts of orange color and the yellow dotted line for human skeleton and five-point inverted pendulum model that includes five purple stars and the red real line for human skeleton. In the test, different moving objects (or different persons) indicated by a five-point inverted pendulum model are shown with different colors.

**Figure 3 sensors-20-00946-f003:**
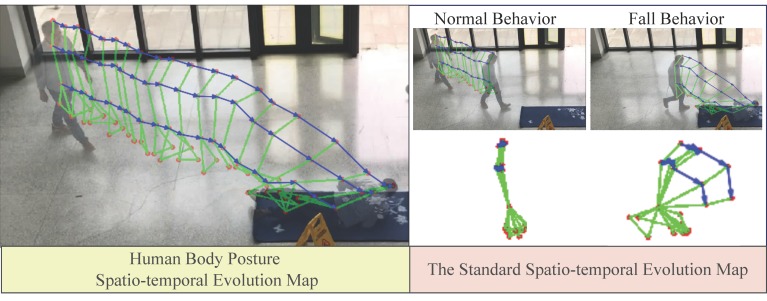
The standard spatio-temporal evolution map based on five-point inverted pendulum model. The left shows spatio-temporal evolution map describes the human body motion at the complete time sequence.

**Figure 4 sensors-20-00946-f004:**
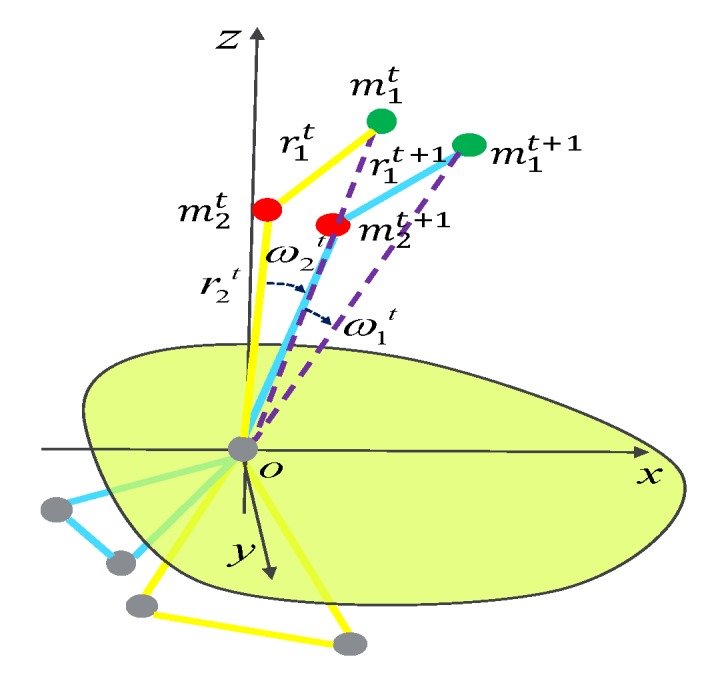
A simple particle system composed of two particles of the end point $EP_{1}$ and end point $EP_{2}$.

**Figure 5 sensors-20-00946-f005:**
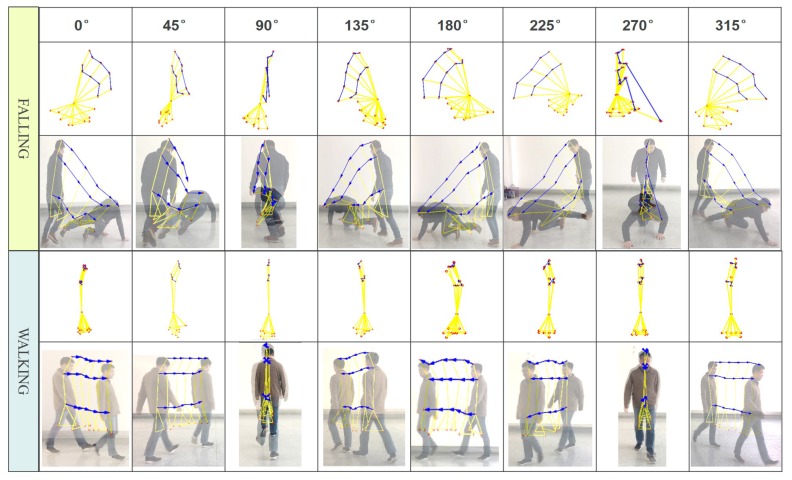
The standard spatio-temporal evolution maps for eight directional falling and walking samples.

**Figure 6 sensors-20-00946-f006:**
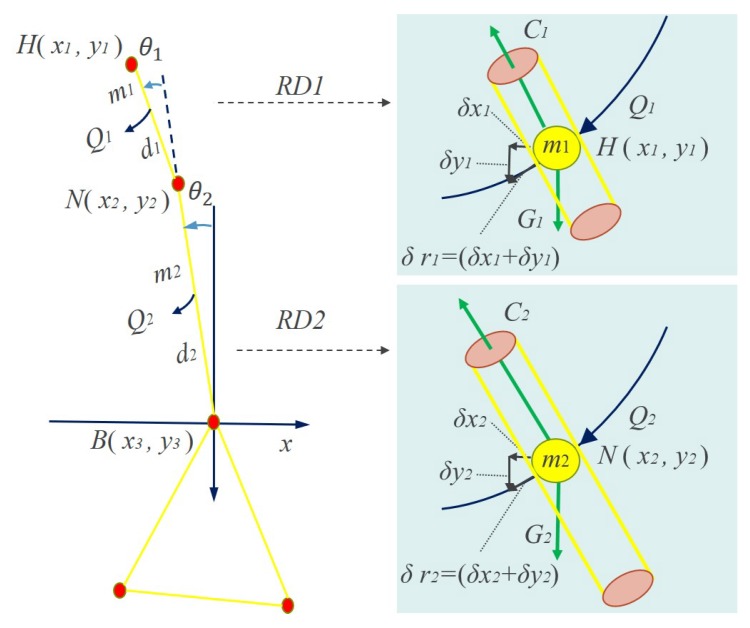
The two-connecting rod system based on five-point inverted pendulum mode. Here, $C_{\left(\cdot \right)}$ is the counter-acting force and $G_{\left(\cdot \right)}$ is the gravity.

**Figure 7 sensors-20-00946-f007:**
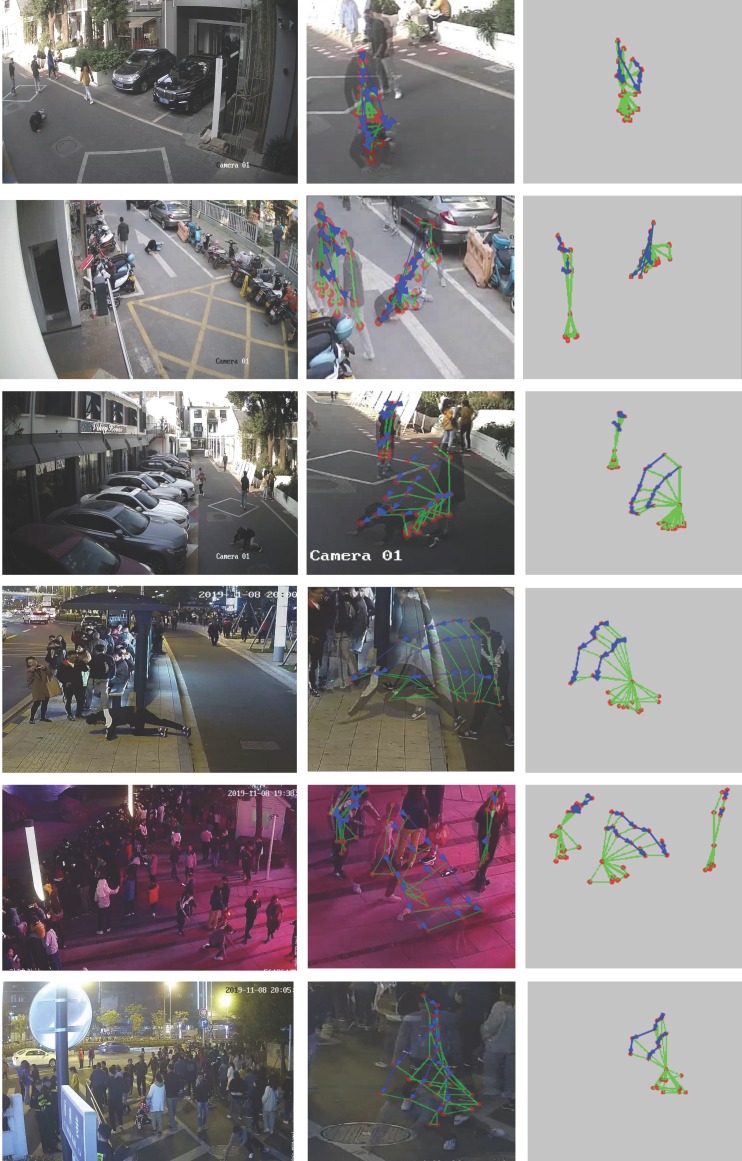
The extraction results of standard spatio-temporal evolution map in (5–10) scenarios. The left are input images from each the scenario; the middle are spatio-temporal evolution maps which describe the human body motion at the time sequence; the right are the standard spatio-temporal evolution maps (if the rotational energy is less than the threshold value, the standard spatio-temporal evolution map does not display for improving the efficiency of algorithms)

**Figure 8 sensors-20-00946-f008:**
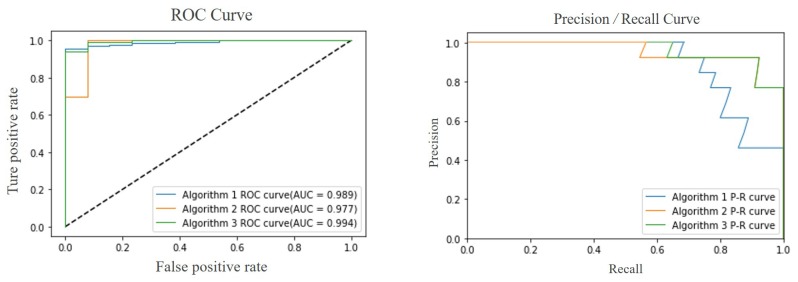
The curves of Receiver Operating Characteristic (ROC) and Precision–Recall (P–R) for three algorithms.

**Figure 9 sensors-20-00946-f009:**
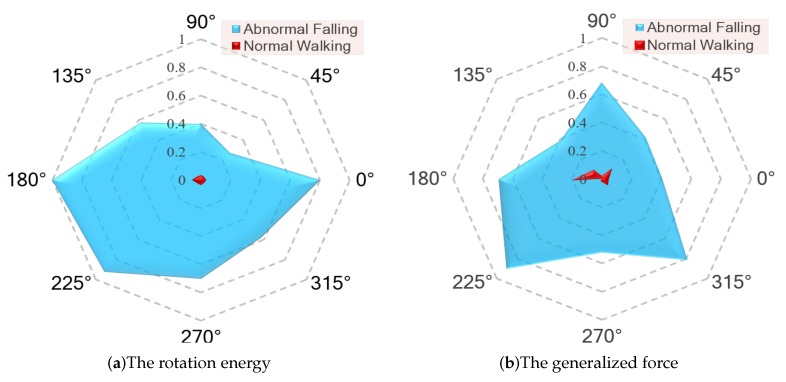
The radar charts of the characteristics in eight directional falling and walking samples.

**Figure 10 sensors-20-00946-f010:**
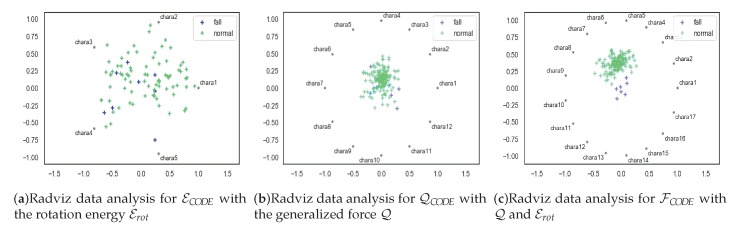
The results of Radviz data analysis.

**Table 1 sensors-20-00946-t001:** The illustration of our video dataset Postures of Fall (PoF)

No.	Scenario	Challenge	Posture of Same Direction	Size
1	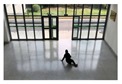	light and shadow	*√*	20 M
2	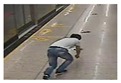	light	*√*	10 M
3	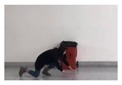	occlusion all directions	*√*	89.4 M
4	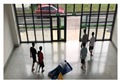	multiple persons	*√*	42.9 M
5	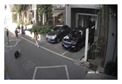	multiple persons	*√*	28 M
6	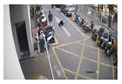	multiple persons	−	17.6 M
7	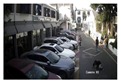	multiple persons	*√*	28.9 M
8	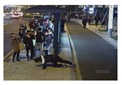	crowded people	−	86.2 M
9	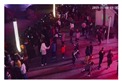	crowded people	*√*	77.8 M
10	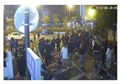	crowded people	−	84.7 M

**Table 2 sensors-20-00946-t002:** The rule of human motion feature tensors

1	2	3	4	*…*	2×T−1	2×T	2×T+1	2×T+2	*…*	3×T−1
$Q_{1}^{t}$	$Q_{2}^{t}$	$Q_{1}^{t+1}$	$Q_{2}^{t+1}$	*…*	$Q_{1}^{T}$	$Q_{2}^{T}$	$E_{\mathrm{rot}}^{t,t+1}$	$E_{\mathrm{rot}}^{t+1,t+2}$	*…*	$E_{rot}^{T-1,T}$

**Table 3 sensors-20-00946-t003:** The comparison of four datasets

Fall Dataset	Scene	Veiw	Object	Action	Sensor	Data
University of Rzeszow Fall Detection (URFD)	indoor (office)	fall–two; ADL–single	single person (designated)	fall + activities of daily living (ADL)	depth and RGB cameras+ accelerator	PNG 16 format and RGB images and accelerometric data
fall Detection Dataset (FDD)	indoor(office ∖home∖ coffee room)	single	single person (designated)	fall + activities of daily living (ADL)	RGB camera	RGB images
Multi-cam video dataset	indoor (office)	8 cameras	single person (designated)	fall + confounding events	RGB camera	RGB images
Postures of Fall (PoF)	indoor (campus corridor∖subway station);outdoor (bus stops∖streets)	single	single or multi-person (random)	fall + walking∖ running∖standing	RGB camera	RGB images

**Table 4 sensors-20-00946-t004:** The evaluation of classifiers for algorithms.

Item	Behavior	Precision	Recall	F1-Score	Accuracy	Average Precision (AP)
ALG1	fall normal	0.90 0.97	0.69 0.99	0.78 0.98	0.971	0.895
ALG2	fall normal	1.00 0.98	0.85 1.00	0.92 0.99	0.958	0.954
ALG3	fall normal	1.00 0.99	0.92 1.00	0.96 1.00	0.979	0.960

**Table 5 sensors-20-00946-t005:** Comparison of our proposed algorithm with four fall detection approaches.

Algorithm	Fall Detection Rate (%)	False Alarm Rate (%)
Shape of the human [17]	90.50	6.70
Approximate ellipse [19]	85.70	20.10
Human Skeleton + ellipse [21]	90.90	6.25
Multivariate Exponentially Weighted Moving Average (MEWMA) + Support Vector Machine (SVM) [23]	93.55	3.34
Our proposed ALG3	**98.70**	**1.05**

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
