# Peer review of "Human Fall Detection Based on Body Posture Spatio-Temporal Evolution"

_sensors, 2020, doi:10.3390/s20030946_

Round 1

Reviewer 1 Report

It is a very interested research paper about Human Fall Detection.

The authors proposed a five-point inverted pendulum mode based on the key points of human body for body posture spatio-tempral evolution.

It is an effective method in the good experimental condition. What about the bad experimental conditions?  For example, the part of target person is hidden in the sth such as people. It may cause that some of the tracked body points are lost.   

What about the camera height and distance? We think the point K and B may become a same point when the camera is set in a higher place or far distance.

It seems that the experimental data are too small.

Reviewer 2 Report

The authors propose an approach for Human Fall detection from videos which is based on spatio-temporal evolution map.

The paper is overall very well written and reads well. The objectives is clear as well as the approaches. The explerimental results somehow support the conclusions even if I think it might be interesting to report and discuss critical conditions where the proposed methods fails. It would add value to the proposal and, moreover, improve the discussion of open issues to fix. It might also impact on the readership interested for this kind of study

Reviewer 3 Report

In this manuscript, the authors proposed a vision-driven methodology for human fall detection. They employed a two-branch multi-stage convolutional neural network to extract the part affinity fields of human skeleton structure and used the extracted features to establish a five-point inverted pendulum model inspired by the robotics. The model can not only effectively detect the changes of body movement but also reduced the computational complexity. Moreover, they also merge the dynamic features from video frames into their model so that the coupled time-space information could be utilized to reveal a human body spatio-temporal evolution map using directed graphs in continuous time. Eventually, they defined theoretical threshold values for quantitative analysis of human fall by solving the rotation energy and the Lagrangian mechanics. A feature tensor using the time-space characteristics of fall instability movement extracted from above methods could be then constructed. It has a great benefit for the readers that the authors simultaneously provide the perspectives of human fall analysis, modeling and feature extraction from deep learning method, theoretical analysis from physical phenomena, and pseudo codes for implementation. The raw data is also available for further checking or studying for participants. Complete and sufficient integrated research results made this manuscript more readable and heuristic. I thought I’m reading a textbook regarding the human fall detection.

Some comments are listed as follows:

Please carefully go through and check the manuscript again. Some typos and redundant words happen. Please define or explain how you get the degrees that listed in the top of Fig. 5. In line 249, page 8, the reviewer recommend use “variational calculation” to replace “differentiation.” In line 295, page 12, the authors mention that they use a binary classifier. Please explain the mathematical characteristics of this classifier and in which process or algorithm the authors used the classifier. Please redefine the phrase “SD posture” mentioned in line 305, page 12.
